# Visual Storytelling, Intergenerational Environmental Justice and Indigenous Sovereignty: Exploring Images and Stories amid a Contested Oil Pipeline Project

**DOI:** 10.3390/ijerph17072362

**Published:** 2020-03-31

**Authors:** Samuel J. Spiegel, Sarah Thomas, Kevin O’Neill, Cassandra Brondgeest, Jen Thomas, Jiovanni Beltran, Terena Hunt, Annalee Yassi

**Affiliations:** 1School of Social and Political Science, University of Edinburgh, EH8 9LD Edinburgh, UK; 2Tsleil-Waututh Nation, 3178 Alder Ct, North Vancouver, BC V7H 2V6, Canada; sthomas.twn@gmail.com (S.T.); koneill@twnation.ca (K.O.); cbrondgeest@twnation.ca (C.B.); jthomas@twnation.ca (J.T.); jiovanni_beltran04@hotmail.com (J.B.); 3Faculty of Health Sciences, Simon Fraser University, Burnaby, BC V5A 1S6, Canada; 4Ravenchild Consulting, North Vancouver, BC V7H 1B3, Canada; terenahunt@me.com; 5School of Population and Public Health, University of British Columbia, Vancouver, BC V6T 1Z4, Canada; annalee.yassi@ubc.ca

**Keywords:** visual storytelling, photovoice, indigenous sovereignty, oil pipeline, Trans Mountain Pipeline, visual geography, environmental health

## Abstract

Visual practices of representing fossil fuel projects are entangled in diverse values and relations that often go underexplored. In Canada, visual media campaigns to aggressively push forward the fossil fuel industry not only relegate to obscurity indigenous values but mask evidence on health impacts as well as the aspirations of those most affected, including indigenous communities whose food sovereignty and stewardship relationship to the land continues to be affronted by oil pipeline expansion. The Tsleil-Waututh Nation, based at the terminal of the Trans Mountain Pipeline in Canada, has been at the forefront of struggles against the pipeline expansion. Contributing to geographical, environmental studies, and public health research grappling with the performativity of images, this article explores stories conveying health, environmental, and intergenerational justice concerns on indigenous territory. Adapting photovoice techniques, elders and youth illustrated how the environment has changed over time; impacts on sovereignty—both food sovereignty and more broadly; concepts of health, well-being and deep cultural connection with water; and visions for future relationships. We explore the importance of an intergenerational lens of connectedness to nature and sustainability, discussing visual storytelling not just as visual counter-narrative (to neocolonial extractivism) but also as an invitation into fundamentally different ways of seeing and interacting.

## 1. Introduction

Visual representations of environments, life, and controversy surrounding oil pipelines take many forms—connecting with emotions, power relations, cultural processes, and diverse values. In 2018 and 2019, a multi-million dollar campaign to aggressively push forwards oil pipeline construction inundated the Canadian public with social media advertisements, billboards and television commercials presenting fossil fuel expansion as clean, attractive and “in the national interest” [1]. As part of efforts to engineer consent to fossil fuel development, a vast array of visual tactics included cartoons of sweet little birds dropping money from the sky into the hands of Canadians [2], graphs suggesting guaranteed economic benefit, and images depicting the building of sparking clean oil pipelines without disruption to surrounding pristine environments. Lacking in the rhetoric and dominant visual representation of controversial pipeline expansion is the depiction of the values, health concerns and stories of those disproportionately affected, and likely to be among the most directly impacted in the future, namely the indigenous communities along the route of the pipeline, whose health, food sovereignty, and crucial stewardship relationship to the land continues to be affronted by fossil fuel developments [3]. Contributing to growing bodies of geographical, environmental studies and public health literature grappling with the performativity of images in environmental contestations [4,5,6,7], this article focuses on visual storytelling of health, environmental, and intergenerational justice concerns of the Tsleil-Waututh Nation (TWN), situated in British Columbia, Canada, at the terminal of the contested Trans Mountain pipeline and tanker expansion (TMX) project. A collaboration between TWN community members and non-indigenous academics in solidarity, we explore images and stories shared in settings where elders and youth alike came to discuss critical community challenges, linking well-being and cultural values with experiences of change and future aspirations. 

The importance of visuals in shaping understanding and sensibilities has long been recognized [8,9,10,11] with the work of Stuart Hall [12] on how visual culture reinforces values, attitudes, and ways of understanding, seminal in this regard. Yet research on visual environmental communication has revealed an increasing tendency toward abstraction or decontextualization of images [9], obscuring the nuanced concerns, desires, and choices as to what is important to visualize from the perspective of different actors [7]. Photography can be a weapon [13] that stigmatizes or victimizes, as well as a tool that empowers; and there has been considerable concern in relation to the use of imagery and visual interventions, generating what some critical scholars have called “crises of representation” [14]. Visuals can also replicate colonialist and imperial traditions whereby photos “from the colonies” were brought back to the imperialist homes to spread the desired image, the camera symbolizing the imperialist’s eye to “accurately” build archives of foreign lands for exploitation [13]. Seeking to counteract this history, critical feminist visual methods have been developed with the aim of situating visual representations within anti-colonial epistemologies and storytelling that can radically reverse the power relation between academics and so-called research “subjects” [15]. A method sometimes embraced in this spirit, “photovoice”, developed originally in public health [16], involves community members leading the processes of visual exploration and has become increasingly popular as a research technique, as well as to promote local solidarity, activism, and deeper understanding more broadly. Notwithstanding the growing body of literature on such techniques, including with indigenous communities [7,17,18,19,20,21,22], there is scant research exploring participatory photography and local visual narratives in the contexts of oil pipeline opposition. Building on debates on the meaning of images and visual storytelling as “more-than-representational” [19] in order to recognize the emotional intensities that are lived, experienced, and conveyed, this article aims to contribute to filling this gap empirically and conceptually, and calls for more research to navigate and explore possibilities, risks, and critical issues in this area. 

The section below contextualizes the photovoice project with a brief discussion of why oil pipeline threats and visual storytelling need to be linked with understandings of the sovereign rights of indigenous peoples. The section thereafter discusses our study methods, designed to support the objective of providing a venue for TWN elders and youth to present experiences and ideas regarding concerns, memories, sustainability issues and values connected to the environment, health and wellbeing, as determined individually and collectively by TWN members. We then explore both the visual stories presented and epistemological issues at play in interpreting key themes and issues that emerged. Finally, we stress the importance of an intergenerational lens of connectedness to nature and sustainability, discussing visual storytelling not just as “counter-narrative” to neocolonial extractivism but also as an invitation into fundamentally different ways of seeing and interacting.

## 2. Background—Colonialism’s Afterlife, Indigenous Rights and Oil Pipeline Threats

The United Nations Declaration on the Rights of Indigenous Peoples (UNDRIP) [23], signed by 144 countries worldwide, includes, among other clauses, the affirmation of indigenous peoples’ right “to maintain and strengthen their distinctive spiritual relationship with their traditionally owned or otherwise occupied and used lands, territories, waters and coastal seas and other resources and to uphold their responsibilities to future generations in this regard.” (Article 25). The notion of territorial sovereignty, however, remains poorly recognized and unevenly applied, as do notions of indigenous sovereignty over governance—usually not valorized by courts and political elites if they clash with existing capitalist extractive interests [24,25]. Linked to these concerns, *visual sovereignty* looms importantly as representing indigenous land, life, and culture by outsiders often risks (subtly or overtly) reifying neo-colonial practices of socio-ecological knowledge production [26]. Braun’s [27] work on “colonialism’s afterlife” on indigenous lands in British Columbia documents diverse ways in which colonial approaches to vision and visuality shape contemporary conversations about land, resources, institutions and environmental imaginaries, from landscape painting to the work of photographers. Researchers studying settler-colonial societies are increasingly voicing the need for work that challenges unreflexive modes of visual representation, to situate visual practices with anticolonial epistemologies and storytelling that oppose “Eurocentric colonial performances of universalization” [15].

Indigenous peoples in Canada have experienced profoundly negative impacts on food sovereignty and related health and wellbeing through processes of colonization, particularly through “the Indian Act” which forced indigenous peoples onto reservations, restricting movement and hindering ability to hunt, fish, gather and access markets for local produce, while timber and mineral resources were extracted from their traditional territories without their consent [28]. Indeed the Indian Act is widely acknowledged by indigenous and non-indigenous scholars alike as “the most colonial piece of legislation imaginable for dominating and controlling every aspect of the lives of the ‘subject nations’ within its territories” [28]. Arthur Manuel, a Secwepemc leader and key strategist in the indigenous movement in Canada, with Grand Chief Derrickson, in their book “*Reconciliation Manifesto*” [28] went on:
...Less than 10 years after the Indian Act was passed, the Canadian successor state was sending troops to the West to attack our peoples and seize our lands, if necessary to starve us into submission…By the time the forces of Anglo-Canadian imperialism were ready to move into British Columbia in the early 1800s, Canadians were so certain that they had broken our people that they did not even bother with formal treaties. They simply pushed us aside and when groups like the Tsilhqot’in resisted, they lured their leaders out of their camps and executed them…. Dispossession was the goal.... Canada was and remains a thoroughly colonial country, built on the dominance of one race over another for the purpose of seizing and occupying their land.

Considerable indigenous-led initiatives have been underway to resist these dynamics with a combination of lawsuits, marches and a variety of protest tactics, including the indigenous youth-led Idle No More movement [29,30]. Yet, despite these efforts to draw attention to the ongoing colonial process in Canada, and the lack of meaningful consultation regarding extractive activities on indigenous land, the TMX project was approved by the Canadian government, involving constructing a 987-km pipeline from the tar sands in the Canadian interior to Canada’s west coast, with the planned expansion to result in a 7-fold increase in marine traffic in the Burrard Inlet [31]—a critical waterway for the Tsleil-Waututh Nation. The main petroleum product to be shipped is diluted bitumen, a blend of oil products that is difficult to clean up because of its heavy sinking properties and long-term persistence in the environment [32]. Exposures to the polycyclic aromatic hydrocarbons (PAHs) in this mixture include potent carcinogens [33,34]. The Canadian government agency responsible for conducting impact assessments for Federal projects has not assessed the likely impact of TMX on the health of the Tsleil-Waututh Nation, despite the nation’s critical location at the pipeline terminus, where, as the “People of the Inlet”, the TWN have continuously, and exclusively, occupied this territory for thousands of years [35]. TWN has never ceded control of its traditional territory to either the British before the creation of Canada or to subsequent Canadian governments, and the Tsleil-Waututh Nation has been contesting the TMX since its inception. 

Traditional foods for TWN include salmon, herring, and shellfish [36], with fishing playing a key role in religious and ceremonial practices, as well as contributing to TWN’s subsistence economy [35]. Consuming traditional foods is a key component of “cultural continuity”—the contemporary preservation of indigenous culture, social cohesion within a community, and self-determination [37,38,39]. Despite health advisories from the Canadian government about marine food contamination and regulations that prohibit commercial harvesting of shellfish [40], many indigenous people still consume these contaminated foods due to lack of nutritional alternatives and the underlying health benefits and cultural values with which they are associated [41]. Oil spills pose a particularly serious risk, as past spills have led to elevated levels of PAHs in shellfish tissues. Shellfish can easily take in carcinogenic PAHs and transfer these exposures to humans who consume them [42,43]. Spilled oil can be trapped under mussel beds for weeks to months and could be readily absorbed and enter food chains [34]. Toxic chemical contamination thus poses a direct health risk to shellfish consumers. Marine biotoxins are also affected by oil spills and dispersants; emerging research suggests an association between oil spills and harmful algae blooms (also known as red tides). Red tides have been reported after oil spills and the use of oil dispersants [44], with observational studies also supporting the contention that oil spills can increase levels of potentially harmful algae [45]. In this context, we set out to explore and deepen the understanding of the meanings of health, environment, sovereignty, and aspirations for the future on the part of the TWN in the context of the pipeline and tanker expansion proposal. We build on the idea that *visual* approaches may potentially encourage both “seeing” as a form of critical “questioning” when contemplating what shapes socio-ecological change (e.g., building further on Thomsen [46]) and seeing as a way of exploring articulations of marginalized knowledges and cultural concerns.

## 3. Methods in Adapting Photovoice

Our approach draws on the premise that cultural values in landscapes can powerfully shape a “sense of being and belonging” [47] (pp.7) and collectively shared feelings of hope, nostalgia, loss, and intergenerational change. This approach dovetails with work on place attachment that highlights various cultural constructions of place meaning and affective and cognitive processes through which the attachment is expressed [48,49]. Exploring the impact of extractivism elsewhere, Askland and Bunn [50] used the term “ontological anxiety” to reflect how extraction can radically transform senses of being, identity, community, and home. Our approach also builds on the work of Castleden and Garvin [17], stressing that photovoice is an iterative process, necessarily open-ended if it is to achieve the desired goals of freeing research paradigms from restrictive and potentially re-colonizing frames. As these scholars argue, pursuing photovoice in community-based participatory indigenous research needs to pay careful attention to what is appropriate and *desired* by participants. Such an approach opens up avenues for narrating cultural, health, environmental and socio-economic threats and changes in a diversity of ways. 

In exploring the usefulness of adapting photovoice methodologies in contexts of struggles over indigenous rights and wellbeing, past scholarship has stressed the importance of embracing indigenous storytelling traditions, ways of knowing and epistemologies [21], including communicating concerns about water and health [51]. In the early phases of planning this project with members of Tsleil-Waututh Nation, elders recommended ensuring a robust *intergenerational* focus in the process bringing together the stories of elders with the aspirations of younger community members. There are profound intergenerational impacts of the structural violence created by Canada’s oppressive policies, including the residential school system [52] and 60s scoop [53], combined with the destructive impact of the loss of food sovereignty on cultural continuity [41,45,54]. Given the traumatic histories and affronts to intergenerational justice, explorations of connection with ancestral values and the making of collective futures, including for future generations,—figured prominently in the co-development of the project methodology.

After the TWN Chief and Council discussed and approved this collaborative project, and following the team’s presentation of photovoice possibilities (lead author SJS presented a diversity of methodological ideas, experiences and permutations first in several team meetings then at an initial community dinner to generate discussion with elders and families), TWN members then offered reflections on photovoice aspirations and adaptations. The protocol was then also approved by the University Research Ethics Board. A longstanding TWN council member (co-author JT) led the recruiting of community participants for this exercise, orienting participants to the project, giving out cameras to those who did not have smart phones and scheduling debriefing sessions at which the photos and stories would be presented for collective discussion. Importantly, researchers purposely provided little direction as to the content of what should be brought forward, leaving it up to participants themselves. Recruitment was by *families*, with families deciding themselves who would take the actual photos, or choose existing photos the family had, which was another adaptation that deviated from the classic photovoice methodology. In some cases, children in the family were the ones who took pictures, and indeed photovoice has been used effectively with children in other settings [55]. In some of our participating families, old photos that had been taken decades earlier and preserved were the ones brought forward.

Three community discussion sessions were held, each beginning with a ceremonial blessing and dinner, gradually shifting to the photos, and then ending approximately three hours later with more informal dialogue particularly around how best to disseminate the knowledge within the community and much more broadly. While in many photovoice sessions worldwide, it is the individual who took the photo who tends to be the main speaker about the photo [16], and other photovoice studies in indigenous communities relied on individual interviews [51], here a more collective approach was taken, as the photos generally represented shared experiences, such that all participants at the sessions had something to say about the issues raised in the photos. Other researchers have commented on the value of centering on the images as a point of conversation rather than the photographers themselves, akin to talking circles [19,55]. In this spirit, additional photos and comments with community members about some of these same themes outside the formal sessions are also incorporated into the discussion below.

In the first session, photos and artifacts that had been sent in by community members or brought by participants were displayed on a table. Alluding to these exhibits, four elder members of the Tsleil-Waututh Nation shared memories and experiences growing up, focusing on changes to the land and water over the years and the resulting impact on food supply, social connections, and long-term health. The session took the form of a discussion prompted by the photos rather than sequential presentations. The second session, held a few weeks later, focused on the photos taken and/or brought by four younger members of TWN, and included not only concerns but aspirations. As was the case in a digital storytelling project conducted by Gislason and colleagues [20], we were interested not only in the knowledge that people conveyed through stories, but also the values displayed and the counter-narratives developed to communicate views about the relationship amongst the environment, health, and sovereignty, and like Gislason’s group, we were also particularly interested in *collective* voices. As such, between the two sessions, photos were posted on the TWN social media network and comments on the photos were contributed by other members of the community. The discussion loosely followed the SHOWeD technique commonly used in photovoice [51,56] adapted in this context for the more collective approach adopted. Not only the individual who brought the photo but the assembled group discussed: what was *S*een in the photo, what was *H*appening at the time, how this photo relates to the *O*verall well-being of the community, *W*hy this situation exists, and what could be *D*one to address this concern. Notes were taken, the sessions were audiotaped and a transcript made of the events. The thematic coding was an iterative process.

Castleden and colleagues [57] discuss diverse views and practices regarding how best to acknowledge the often-profound intellectual contributions of participants in community-based participatory research with indigenous peoples, stressing the benefit of shared authorship in working towards a common goal of respecting indigenous knowledge both within the academy and the community. Other scholars have also discussed co-authorship as a way of recognizing community participants as co-producers of knowledge [58]. Community-based team members who played a key role in shaping the study design and its implementation from the early stages served as co-authors along with the academic authors; two photovoice participants later joined the authorship team. Dynamics of knowledge production, whose voices guide knowledge shared and the medium itself (recognizing the need to avoid prioritizing written modalities over the more traditional oral modalities), were reflexively discussed at various stages during and between the photovoice events. At the third photovoice discussion session, the three generations of participants from the previous two sessions came together for a further community dinner event, in some cases bringing family members and friends. Enlargements of a dozen selected photos were mounted along with quotes and a draft of this article circulated to those present, with each person’s draft highlighting their own words and photos, or in one case, the photos in which they appear. Participants elaborated on the reflections offered previously, suggesting edits and additional detail. A rich discussion then occurred about possible follow-up activities.

## 4. Findings: Visual Stories, Intergenerational Dialogues, and Sovereignty Aspirations

Inter-related themes characterizing the visual storytelling included current concerns about the health of the Nation; the environment and how it has changed over time; sovereignty—both with respect to food sovereignty and more broadly; the overall concept of well-being and particularly the role of water in this regard; and the vision for the future. The intergenerational impacts constituted a strong through-line within the highly integrated approach to the environment, health, cultural continuity, and sovereignty aspirations depicted in the visuals and stories they provoked, as discussed below.

### 4.1. “The Water That Heals and Teaches Us Is Now so Polluted by Colonization”

Water has always been integral to the health and well-being of indigenous peoples, crucial not only to meet physiological needs but also for cultural ceremonies, social activities, and traditional ways of life [51,59,60,61]. Water is a particularly important theme for the TWN, as noted by a member of the Youth Council who participated in the photovoice exercise (co-author KO). Offering documentation from his own experience as well as seven formal interviews he previously conducted with community leaders [61], he cited, among others, the highly respected elder known as the nation’s “Ta-ah”, Amy George, who taught him that the first grandmother of Tsleil-Waututh “*came up from the* *water*”. Water was a major focus in many of the photos brought forward, see Figure 1, with KO explaining the importance of canoeing:
I got back into the canoe because of the ties to our ancestors. In the canoe, you are pulling and more connected to the land, to the water and the creator…When I look at the picture I see wellness. When you are connected physically, culturally and spiritually you are well… It takes several people to carry the canoes, they are heavy. The first step before getting into the canoe is to see how well you work together. You need to have balance… It’s the power behind the canoe that heals. As you pull, the power is with you…The teachings [from the water] have to be experienced. For some, canoeing is a big part of growing up. More people are trying to connect with the ancestors. The canoe is a pathway to that.

He then went on to explain how he heard that the color of the water has changed and he himself noted that he could feel the silk of oil on his skin. “*If you don’t shower right away you get rashes*…*The water that heals and teaches you is so polluted by colonization*.”

A participant, who was originally from another indigenous nation 400 km away (co-author TH), explained how her connection to her home is through the water. When she gets lonely for home, she goes to the water and imagines it flowing all the way home and back. She explained that they eat similar food in her home community but she won’t eat it at TWN. “*It’s not* *safe*,” she explains:
I think it is sad, you desperately want to connect with something that your people have connected to since the beginning of time. I call it blood memory. Until those pieces of ourselves that have been harmed through colonization are filled, the longing will continue…People don’t think of oysters as food here anymore. There are different opinions regarding what is or isn’t safe. You can’t prove the food has not caused cancer.... We have a reciprocal relationship [with water] as a living being. The signs are telling us she is being harmed... It [the contamination] is against our spiritual relationship with Mother Earth…

Another member (co-author JT) shared that her dad canoed since he was ten-years-old; she proudly noted that he was a champion for many years and traveled around competing against other nations. Her photo also showed a canoe (Figure 2), as she explained:
The increase in tanker traffic will have a huge impact on canoeing. This is a huge part of our culture and we practice every day. Tanker traffic increases the danger. If there are too many tankers we may need to stop canoeing. Even if we put aside the cultural aspect of canoeing and the canoe club, the water itself will not be ok if tanker traffic increases. What they don’t talk about is that each tanker is allowed to spill up to 100 litres without having to report it as a spill. If you have more tankers the impact will be huge...

In addition to the content of the narratives, the shared experiences and even the similarity of the visuals chosen, provided a clear counter to the dominant narrative promoting oil pipelines and tanker traffic.

### 4.2. “The Shipping Has Killed Off Much of the Natural Habitat”

Inextricably linked to concerns about the impact on human health were environmental concerns, both of local water pollution as well as those attributable to climate change, including impacts on red tides and high tides more generally. An elder explained how the rivers had tributaries where they used to fish, but the river was diverted when an industrial park was built. “*When the tide goes you can see a big lake. It was drilled by the sawmill to be a tailing pond so they could still work when the tides went out. The shipping has killed off much of the natural habitat*.” He noted that although freighters are not allowed to dump their ballasts within 12 miles, the red discoloration of the water related to freighter traffic is still happening.

In exploring the concept of “solastalgia”—the distress caused by the transformation and degradation of one’s home environment [62]—scholars have called for a better understanding of indigenous peoples’ lived experiences of water and landscape transformation and environmental degradation [63]. Chandler and Lalonde highlighted the importance of cultural continuity by showing that indigenous communities that preserve their heritage culture and take steps to control their own destinies, are much more successful in decreasing the risk of youth suicide [64]. In this photovoice exercise, the profound hurt felt by degrading the natural habitat and the water systems (Figure 3) was explicit in both discussion groups. One of the younger participants stated:
We are connected by our love of the land, for she is a living being, a relative. Ownership of the land was a foreign concept to First Nations people because we knew we could not own a living being, Mother Nature. Our relationship with her was always a reciprocal one. Let us make sure we keep it that way.

Another elder explained how TWN children all used to swim there but they cannot anymore as the tides are too high and getting higher every year. “*There was a huge flat rock in front of their place where you could step off the bank and onto the rock. Now it is out in the water. It is all changing*.” Others noted how large trees are falling into the water due to soil erosion. One elder reported that: “*[ancestral] remains were found where the bank eroded—they think it was three different people. They did a ceremony and placed them up the bank.*”

The focus on preserving the sanctity of, and continuity with, previous generations underscored the transgenerational nature of the profound social-cultural-environmental nexus articulated throughout the sessions.

### 4.3. “What Used to Feed Families Is Now Causing Them to Get Sick and Die”

One of the elders shared artifacts found on the beach over many decades, including arrowheads, beads, tools and other items collected by her mother, herself and her granddaughter. She noted that one of her most treasured items, a crystal-like piece, was more than 4000 years old, according to a local archaeologist. Explaining that different items were used for different purposes, often related to hunting marine life, she stressed how her family used to live off the beach harvesting and eating the crabs, clams and other bounty (Figure 4 and Figure 5). “*Now, many of them are dealing with cancer, many different types of* *cancer*”, she explained, deeply lamenting that it is no longer safe to eat the food from the beach and this has been a major blow to the community.

Another elder at the photovoice session explained how he saw his first red tide in 1951 and there another did not occur until 1958, “*then it was once a year in the seventies. Now, if it is a hot summer there can be three red tides*”. He lamented that he used to take the children swimming but now it is too dangerous, not only because of the high tides but because of the water quality; he explained how he scratched himself on barnacles and the wound quickly became infected, such that he stopped harvesting clams in the mid-70s as he “*does not trust the water*” anymore. Continuing the conversation yet another elder explained that he and his gang used to go fishing for tommy cod, “*there were so many fish you could practically scoop them into the boat*. *It was like they were trying to jump in*.” He said that his grandson told him he was going to be a fisherman and when asked why, he responded that his whole family were fishermen. Pursuing this theme further, a young participant talked about how she used to dig clams with her grandmother; “*they used to soak them in a bucket of fresh water with rolled oats to get them to spit out the sand*” as a means to clean them. To underline the importance of seafood to the TWN diet, another participant reminded the researchers: “*We had a saying here: When the tide goes out, the table is set*”. Yet another, who continued to harvest crabs, explained:
My father taught me to harvest crabs, I taught my son, and he will teach his son. This is our way of life. We used to provide crabs for the community; now most families won’t eat these crabs. But DFO [Department of Fisheries and Oceans] hasn’t told us they are unsafe to eat.

The concern about cancer was omnipresent. Indeed, other indigenous communities downstream of the Alberta tar sands have also expressed considerable concern about elevated cancer risks [65], albeit scientific studies have been limited by methodological challenges. One participant, co-author JT, shared her reason for participating in the photovoice exercise:
I kind of know some health history with two families, and these families are the ones who lived off our beach for years and years, and didn’t stop when we were told to stop eating down there. … In one family there was one, two, three, four of them that have cancer. There’s only one of them left of the four of them. The other family is the two guys that were out. Their sister - she never stopped living off the beach and she suffered with cancer for years and years. …We can’t prove that it was the beach - but I want to ensure that their stories are told and never forgotten.

### 4.4. “We Were the Ones Who Watched Over the Land. Now We Feel That the Refinery Is Watching Us.”

The distrust and strained relationship with government authorities figured prominently in the stories. Even having to check with Canadian or provincial government health authorities on the safety of food and water was seen as oppressive and undermining indigenous sovereignty. As one participant explained:
Having to go to get it [the eating of harvested shellfish] cleared by some official lab --It’s a form of oppression. - … It’s kind of like getting permission, you know. Even if it’s permission from a health perspective. And I guess to make matters worse, the science is probably less clear than optimal as to what’s safe and what’s not.

An elder told how they used to go out in the boat, shoot ducks and bring them home. He recounted how they would take about three each. A member of the younger generation noted that her generation also hunted ducks. “*Then the cops would come… no more duck hunting*.” Another elder explained how many artifacts were taken [by settlers] when the highway was built; the workers came across a lot of bones, and he was quick to add: “*You don’t disclose where the bones are because you don’t want people [settlers] to steal them*”. He went on to explain how they “*used to have a lookout where the refinery is. There were lookouts all the way up so you could send signals. Signals would be sent back, to let everyone know if someone was coming through*”. Another elder acknowledged that when she was a young girl she wanted to blow up the refinery, but her dad told her she “*would blow-up her community too, so it wasn’t a good idea.*” Today, she noted, they still see the flame [from the refinery] (Figure 6) and “*feel it as an invasion*”. And she notes that members will post photos on Facebook when the flame gets too large. Simpson [66] discussed how the history of fossil fuel extraction “served a purpose even more immediate than capitalist accumulation—it served to consolidate a nascent settler-colonial state’s claims to authority over territory”. The photos and discussion vividly conveyed this sentiment, particularly accentuating how the new controversial TMX project is layered on histories of invasion and territorial ordering built on the (neo)colonial logic of promoting oil above all else.

### 4.5. “Unsustainable Technologies Will Be Gone; Our Nation Will Still Be Here.”

Counter-narratives to extractivism can reflect a range of perspectives and have the potential to assert voices that disrupt dominant ideologies [20]. A powerful counter-narrative explicitly emerged when a youth (co-author JB) started the second session by sharing his photo (Figure 7) that, to him, demonstrated “contrast”. He explained how it was composed of the TWN children’s daycare centre with the solar panels in his photo representing:
The future’s embrace of sustainable technology versus the outdated clutch to fossil fuels as the main means of energy creation, sustainable technology. “A point of pride for my nation is our drive toward self-sufficiency and our investments towards cleaner self-sustaining technology. Our solar panels and wind towers are steps in the marathon that is our journey.

His words resonated with the interviews conducted previously by O’Neil [61] in which interviewees stressed: “*Tsleil-Waututh Nation is not anti-development, we are anti-pipeline, we want to bring the inlet and fish back to* *life*.” In referring back to his photo, this participant indicated how across the water one can see the oil refinery, representing old unsustainable technology. He explained how the daycare speaks to the future and the responsibility TWN members feel to children, grandchildren, and generations to come. He expressed the belief that the best way to reach people is through storytelling. Pointing to the house in the photo he also explained:
When I think about being from TWN - it is about community. When we have funerals, people know their roles; when we have community events, everyone has a role. It is common to have multigenerational homes and be located close to each other. I don’t think non-members have an understanding of what it means to be First Nations. There was a period of time - maybe 20 years- when TWN didn’t have the capacity or resources to build homes for members. Once that changed we were able to bring people back when new homes were built. …Some say if we take too much land for housing we won’t have the forest to connect to. I don’t agree. Anyone who comes back here feels welcomed back home. The land is where we live but home is the people.

The words of this TWN youth leader reflected the resurgence of indigenous youth across many indigenous communities in repatriating severed relationships to nature and traditional practices linked to holistic concepts of health [67], and in which stewardship is central to resilience and health [19]. So too was the notion of gender equity. In reference to gender discrimination in the highly-colonial Indian Act in Canada, this TWN youth went on to state: “*We have [women] elders who grew up here, but because they married non-members they had to leave the community. There are a lot of elders who feel they missed out because they had to leave. Mostly women*” and he stressed “*This is the place where I would like to raise my family*.” In the final discussion, he poignantly noted:
*Our people look down the line seven generations to make sure it will be ok for them. We need to make sure the pipeline doesn’t happen.… We need to educate the settler population so they have a basic understanding of First Nations’ connection to the land and spirits to the water…. There is no way to contain or fix an oil spill. What do we do to stop this, besides a violent protest? …We need people to realize that if this [the TMX] goes through and it goes wrong we are all hooped…My photo emphasizes the contrast. The futile lifestyle is the plant across the water [pointing to the oil refinery]. We have been here for generations and the earth did well in our care. …We traveled to New York a few years ago to talk to companies connected to Kinder Morgan [the owners of the pipeline before the Canadian government bought it]. They were shocked by what we told them*.

The choice of framing thus highlighted the contrast between an indigenous vision for future development built on care, community and sustainability, one on hand, and the ongoing yet moribund fossil fuel development, on the other, with divergent timescales at play.

## 5. Discussion

Efforts to challenge notions of a “detached” and “objective” gaze [68,69] are resulting in growing critiques of dominant representational practices. While deceit has long characterized the colonial project, the plethora of visual imagery bombarding the airways to promote fossil fuel expansion in Canada constitutes part of the contemporary “post-truth” world where facts and science are routinely subordinated to the economic and political interests of elite actors, facilitated by advertising laws that permit *non-factual* political or government messaging [2]. Despite the Canadian Federal Government’s self-promotion as a climate leader, and official acknowledgement of the “climate crisis” generated by fossil fuels [70,71], the production of highly polluting fuels has increased sharply in Canada, leading to Canada becoming one the world’s largest oil producers. Misleading evocative visual imagery promoting pipeline expansion is likely to increase considerably according to the recently elected conservative government in Alberta [2,72]. While photos have indeed surfaced of environmental degradation, spills and oil company infractions, taken by citizen-turned-photographers using imagery to demonstrate failures of corporate so-called “environmental stewardship”, and the public media has occasionally also shown photos from struggles to resist pipelines, including indigenous and non-indigenous people from various movements marching together, images depicting experiences and aspirations of indigenous communities, such as discussed in this article, have been almost completely absent from public consideration. The early developers of ‘photovoice’ methodologies articulated three main goals around this technique [16]: (1) provide stimulus for active dialogue within a community; (2) create a safe environment for critical reflection, and; (3) promote social change, through informing the wider public and policymakers. Building on rich indigenous oral storytelling traditions, photovoice can not only provide a vital set of counter-narratives to extractivist agendas and neo-colonial framings, but potentially also offer avenues for embracing multipronged ways of seeing the impacts of fossil fuel expansion and offer fundamentally different ways of envisioning community futures.

Photovoice has been used previously with indigenous communities to document land and water resource management, food security, and health issues [19,51], and as consistent with the theoretical underpinnings of this technique, is flexible and adaptable to the context in question. The extensive use of photovoice in indigenous communities led some researchers, working with Anishinaabek (in Ontario, Canada), to articulate four key components, namely, including a ceremonial opening, a process of group sharing at learning circles, sharing a meal at each sharing circle, and participant-established group guidelines [73]. The Tsleil-Waututh adapted methodology profiled above builds further on these components to accentuate the importance of adopting an inter-generational perspective to photovoice processes. While considerable literature is accumulating on the gendered dimension that lends itself to capture this technique [7], and others have noted how stories can surface the importance of generational ties and the “distinct sense of place“ that is “passed inter-generationally” [20], much less discussion has been afforded to intergenerational aspirations articulated through photovoice work. In the Tsleil-Waututh photovoice project, the intergenerational dimension figured prominently, both in terms of participation and as a focus of concerns articulated. Indeed, all the narratives and photos shared by participants reflected a multi-generational perspective consistent with indigenous cultural traditions and fluid, interconnected visions of time.

Our analysis resonates with the recently published article in this journal [74] by Larson and colleagues, in which 190 indigenous Australians across four communities were interviewed to ascertain impacts on the well-being of land and sea management initiatives. In common with indigenous peoples in Canada and elsewhere, the authors documented how traditional land and water management practices in Australia involve not only maintaining the physical environment but also carefully nurturing values, stories, and cultural obligations associated with the region, with the need for autonomy in creating opportunities for development. In a study of the anti-pipeline movement in Canada several years ago [75], researchers observed that the intensifying conflict not only reflects widespread public concern about serious risks to water, land, and climate but also escalates tensions between the federal government and First Nations regarding treaty and rights violations. Indigenous communities in Canada continue to encounter infringements on indigenous lands by extractive industries and developers, with dispossession and colonialism largely unabated. Increasingly indigenous and non-indigenous scholars and activists alike are calling for radical transformation of the political, economic and thought systems driving climate disasters, arguing that framing the climate crisis mainly in terms of greenhouse gas emissions risks obscuring the racial, gender, and social class injustices that need to be addressed in a just transition [75,76]. Yet few palpably experience the profound impacts of pipeline expansion as do indigenous peoples.

Stories are crucial to social change; moreover, storytelling has always been a major part of indigenous cultures, including that of the TWN [61]. As other scholars stated: “stories emerge that offer much-needed counter-narratives to the mainstream discourse … about the climate and social inequality … These deeply grounded understandings, stories, and visions of the problems and solutions are crucial to bringing about the kinds of transformations necessary” [75]. Storytelling that privileges lived experiences can serve a myriad of purposes ranging from surfacing subjugated knowledge [77] to transmitting knowledge across generations and “conveying tacit ontological assumptions” [78]. Nonetheless, community-relevant actions and change must continue to underscore the goals of visual methods of inquiry [18,19]. Many photovoice projects include public viewings of the photos/captions to facilitate engagement of other community members, government agencies and/or researchers to better understand the values, concerns, and aspirations that may escape textual descriptions derived through interviews and documentary analysis [7,19,55]. The project described here is now entering its phase of the wider community, policymaker and public engagement, expected to be an ongoing process over the coming years, using all the above techniques plus social media and taking on a life of its own as the struggle against pipeline and tanker traffic expansion continues. Follow-up activities suggested at the third community dinner included establishing a permanent display of photos along with audiotaped interviews with the photographers, the creation of a documentary video, online interactive stories, and a larger event with policy-makers and the general public. One suggestion was a wider invitation for photo contributions with a weekly social media posting of a single photo and invitation to the community to comment. Additionally, a follow-up to the stories presented in the photovoice sharing has already been launched in 2020 in the form of a toxicology study to measure biotoxins and chemical contaminants in shellfish harvested by TWN members along with a community health study, as protests continue and the legal case against TMX proceeds through the courts. Photovoice may fundamentally be best understood not as a time-bound finite exercise but rather part of a larger process of multiple methods of contesting the expansion of fossil fuel extraction and nurturing alternative community futures, rooted in efforts “to take seriously the conceptual and empirical contributions of indigenous epistemologies” [79].

## 6. Conclusions

Expansion of oil tanker traffic in this area, as proposed by the TMX project, represents much more than a risk to the *physical* health of the Tsleil-Waututh people, but also to the *mental* and *spiritual* well-being associated with the disruption of important traditional practices connected to the water itself. Such notions can be viscerally illuminated through visual storytelling in ways that may be obscured by other research methods. The contrast between future aspirations of Tsleil-Waututh and the fossil fuel industry was vividly conveyed through visual storytelling, creating a counternarrative to the uncritical decontextualized images that dominate the public purview. Given the important role of *visuals* in shaping understandings, visual storytelling can be particularly powerful in elucidating deep cultural dimensions and lived experiences, in addition to enhancing understanding of profound ongoing health and environmental concerns. An *intergenerational* justice lens can be particularly important in this regard. While visual storytelling is never straightforward, such techniques can invite vastly new ways of seeing, photos are “evidence” but also conversation-openers with more than representational qualities given potential intergenerational importance and epistemologies that can critically challenge colonial extractivism with alternative approaches to interacting. In attempting to inform measures in line with UNDRIP and planetary survival, researchers and the public at large may find value in paying careful attention to relational processes that undermine indigenous sovereignty, including *visual* sovereignty. Such considerations need to loom large in intercultural research collaborations and actions using visuals to pursue transformative goals.

## Figures and Tables

**Figure 1 ijerph-17-02362-f001:**
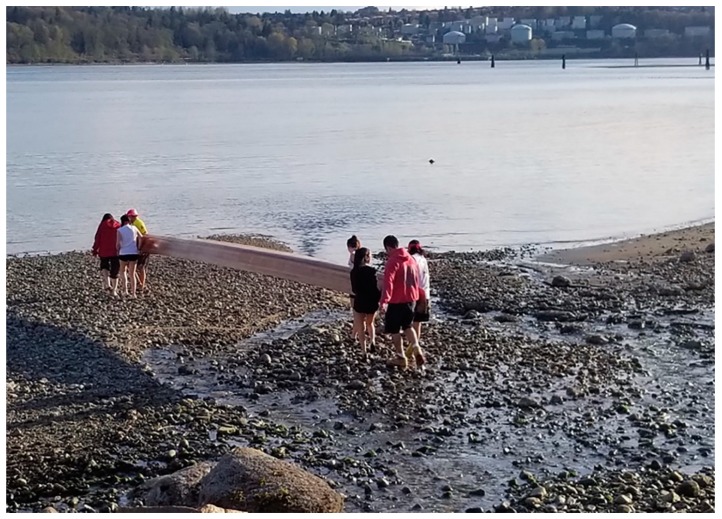
The canoe photo (Photo Credit: Kevin O’Neill).

**Figure 2 ijerph-17-02362-f002:**
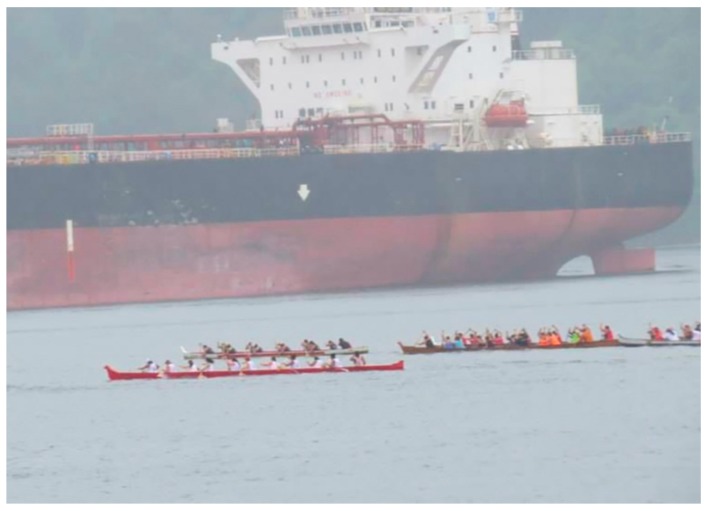
Showing a canoe beside a tanker (Photo Credit: Jen Thomas).

**Figure 3 ijerph-17-02362-f003:**
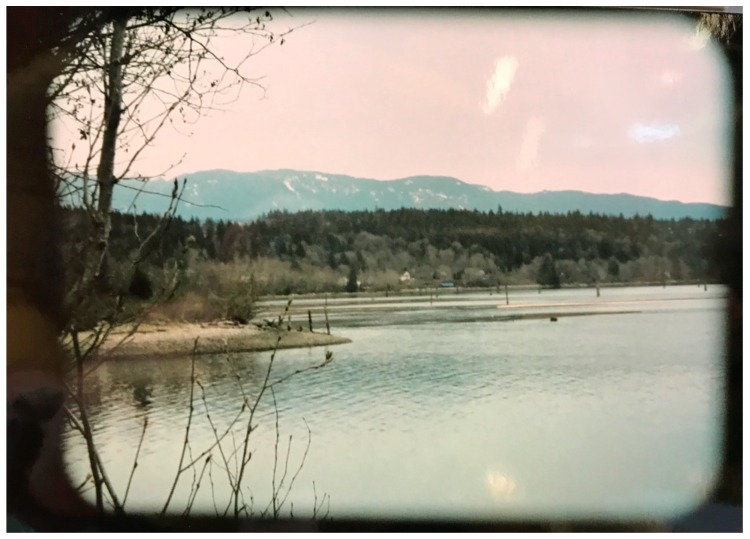
(Photo Credit: Lorelai Thomas).

**Figure 4 ijerph-17-02362-f004:**
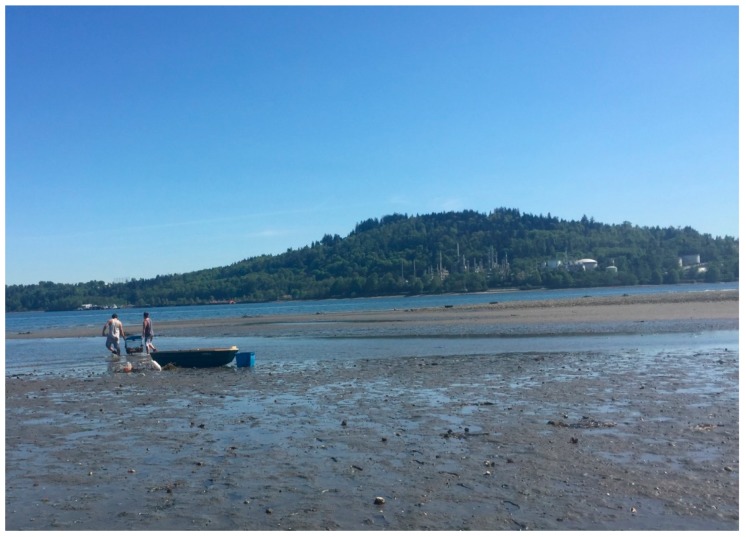
Crab fishers (Photo Credit: Jen Thomas).

**Figure 5 ijerph-17-02362-f005:**
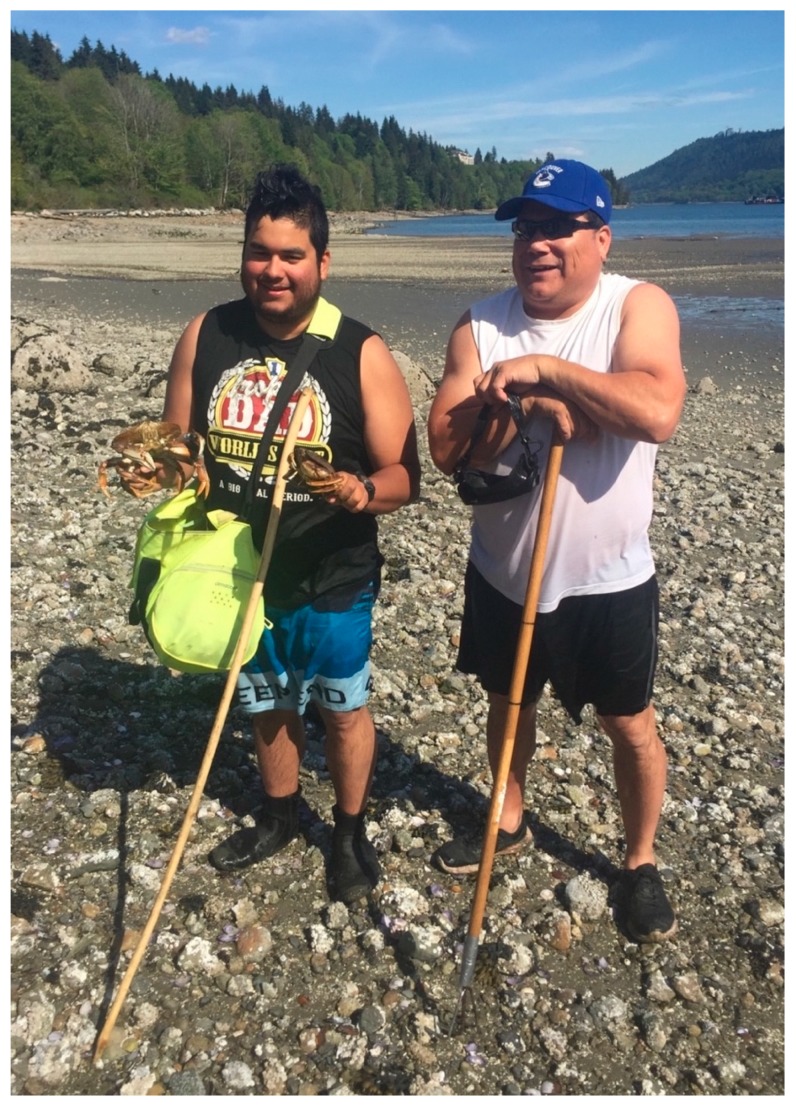
Crab fishers (Photo Credit: Jerry Spiegel).

**Figure 6 ijerph-17-02362-f006:**
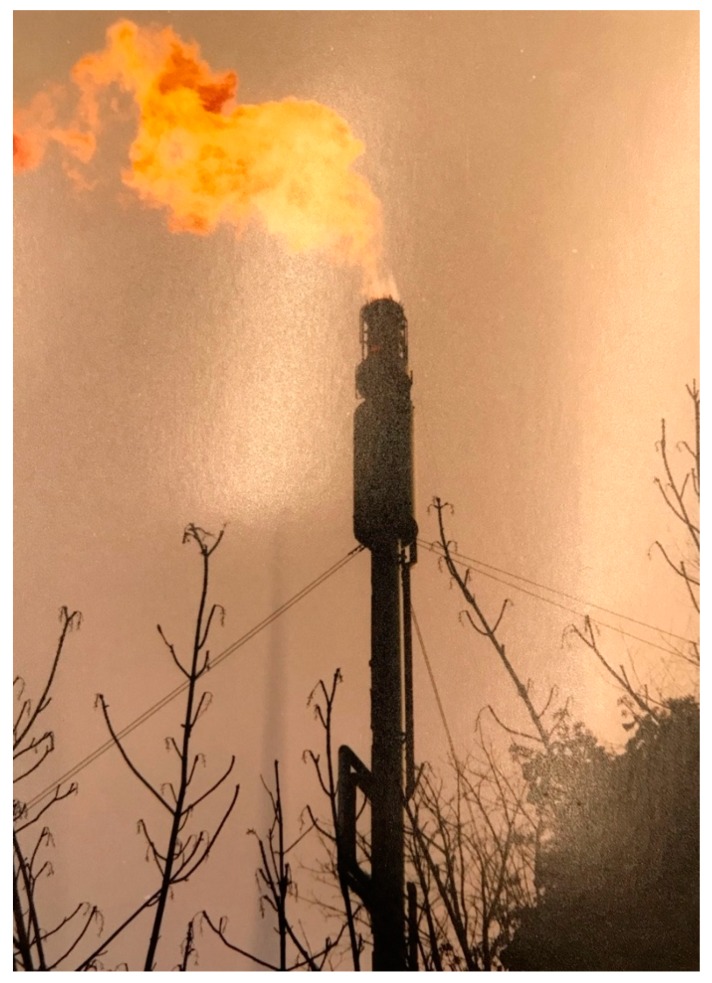
Of the flame on the refinery (Photo Credit: Lorelai Thomas).

**Figure 7 ijerph-17-02362-f007:**
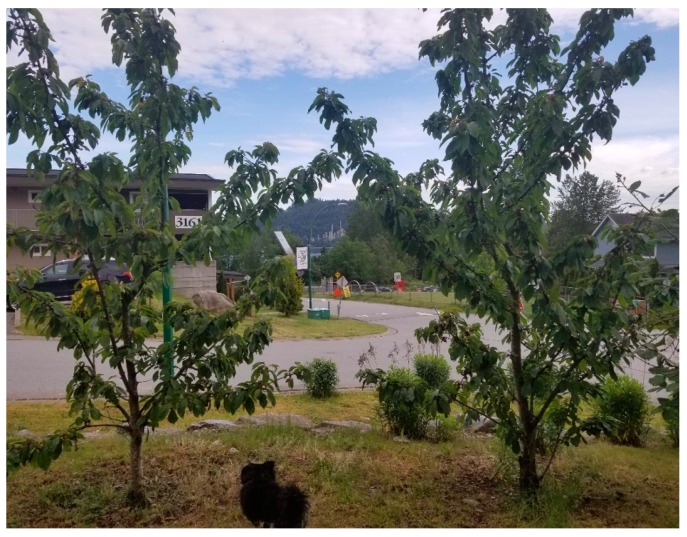
(Photo Credit: Jiovanni Beltran).

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
