# Peer review of "Visual Storytelling, Intergenerational Environmental Justice and Indigenous Sovereignty: Exploring Images and Stories amid a Contested Oil Pipeline Project"

_ijerph, 2020, doi:10.3390/ijerph17072362_

Round 1

Reviewer 1 Report

This article discusses collaborative research between Tsleil-Waututh Nation community members and non-Indigenous academics in using visual storytelling to examine the community response to the highly contested Trans Mountain Pipeline. The research exemplifies the way in which photovoice, as a form of visual Indigenous storying, examines the lived experiences of participants. The stories portrayed themes of Indigenous wellbeing related to connectedness to land and water. The resulting narratives counter media campaigns promoting the oil industry.

The article is well-organized and contains the standard components. Each section is well developed and the literature is synthesized throughout the article. The research answers the question of how visual storytelling can be used to investigate the health, environmental, and justice concerns of a First Nation.

The research methodology is clear, including the co-development of the research, the use of photovoice as visual Indigenous storytelling, the importance of intergenerational knowledge, and the collaborative nature of the data analysis.

The research is important in the field of Indigenous health as it demonstrates a way researchers can respond to the self-determined research needs of Indigenous communities. Collaboratively, Indigenous community members and researchers can examine the lived experiences that impact health. 

Please note the following:

There is an inconsistency in spelling of last name: Castledon (line 173) and Castleden (line 594)

Line 326-327 needs editing

Line 333-334 needs editing

Author Response

Thank you very much for these nice comments and the thorough review. We corrected the typo (they both should have been Casteden), and edited the line in question.

Reviewer 2 Report

The authors of ‘Visual Storytelling, Intergenerational Environmental Justice and Indigenous Sovereignty: Exploring Images and Stories Amid a Contested Oil Pipeline Project’ have produced an important manuscript which makes a highly relevant, incisive and timely contribution to advancing our understanding of the current moment in Canadian history.

This article presents a body of work developed through a collaboration between settler researchers and members of the Tsleil-Waututh Nation, which explores Indigenous experiences and visual depictions of the impacts of the Trans Mountain Pipeline project on the physical, mental and spiritual health and wellbeing of this nation. Importantly, this work places current events in the context of settler colonialism in Canada and effectively uses a critical theoretical lens to describe the continuities between past and present colonial projects, with particular attention to role that resource extraction has played in the Canadian colonial project. In addition, the authors offer a novel methodological approach, PhotoVoice, and describe how it has been adapted through this project to reflect the needs and worldviews of the Tsleil-Waututh knowledge holders, imagine producers and collective storytellers. This alone makes a valuable contribution to the literature.

This article is particularly effective as it foregrounds and integrates issues of visual storytelling, Indigenous sovereignty, intensive resource extraction and socio-ecological health as well as grounds this work in place and brings it to life through concrete images and examples, particularly centred around food, land, health and cultural continuity. Finally, this article embodies the recommendations it also makes and that is to identify, challenge and disrupt misinformation campaigns and political agendas which mask evidence and bury the voices and living experiences of populations. The authors do this by presenting a body of research which foregrounds the voices and experiences of the people who are being impacted and offering up a new kind of body of evidence which challenges current extractive ‘business as usual’ practices.

Overall, this is a thoroughly researched, well written and polished piece of work and is, in my view, ready for publication upon receiving a final copy edit. I can see that this work will be useful for researchers, educators and health practitioners and will also provide a useful elaboration for methodologists.

Author Response

We very much appreciate this complimentary review.  We too hope that it will be useful for researchers, educators, health practitioners and methodologist. Thanks again.

Reviewer 3 Report

The paper describes a photovoice initiative undertaken by the authors to study the health, social, and cultural impacts that an extensive oil pipeline might be having for The Tsleil-Wautuh Nation, an indigenous community living next to the pipeline. To do so, they conducted a total of 3 photovoice sessions with the elders and youngers of such a community. The paper is well written, interesting for the reader and may result in a contribution to the ongoing literature on the health and social impacts of controversial facilities and infrastructures. However, and despite its merits, I find an issue with the paper that makes me doubt on its suitability for IJERPH. According to the text, it seems that a remarkable proportion of the actual participants of the photovoice session are also the co-authors of the manuscript/study. This situation would not be easily justifiable in other areas of the Social Sciences and the Hummanities. 

Regarding the content of the paper itself I would recommend the author to include as figures some of the pictures they say were used as visual tactics to foster support for the pipeline within Canadian society. This could help the reader to see which are the visual products and initiatives that should be contested to secure TWN rights and health. 

Section 3 could also be enriched by mentioning the literature on Sense of Place and Place attachment, which is a rich field in environmental psychology and related sciences. Indeed, the construct therein called "place attachment disruption" specifically refers to the negative psychological and community impacts that changes in places beloved for an individual or a group may imply. Some key references would be Lewicka 2011, Brown and Perkins 1992 or Scannel and Gifford 2010.

Author Response

We thank the reviewer for the comments about the strengths of this paper, and respond here to the three concerns raised.

First, we thank you for drawing attention to the fact that 4 of the 8 co-authors were participants in the photovoice exercise. Although the role of each author is listed in the “author contribution” section, we had not specifically commented in the article itself as to why this occurred. As such, we have now revised the article to be clearer that authorship reflects the co-production of knowledge, not only in designing and implementing the project but also in the data synthesis and interpretation. We have now discussed and Castleden’s work precisely on the issue of co-authorship in community-based participatory work in Indigenous communities (ref 57), and an article about “Walking the Talk…” (ref 58) about co-authorship in community-based action research. Importantly, we also added considerable text to the last paragraph in the Methodology section. This paragraph now reads as follows:

Castleden and colleagues [57] discuss diverse views and practices regarding how best to acknowledge the often-profound intellectual contributions of participants in community-based participatory research with Indigenous Peoples, stressing the benefit of shared authorship in working towards a common goal of respecting Indigenous Knowledge both within the academy and the community. Other scholars have also discussed co-authorship as a way of recognizing community participants as co-producers of knowledge [58]. Community-based team members who played a key role in shaping the study design and its implementation from the early stages served as co-authors along with the academic authors; two photovoice participants later joined the authorship team. Dynamics of knowledge production - whose voices guide knowledge shared and the medium itself (recognising the need to avoid prioritising written modalities over the more traditional oral modalities) were reflexively discussed at various stages during and between the photovoice sessions. At the third photovoice discussion session, the three generations of participants from the previous two sessions came together for a further community dinner event, in some cases bringing family members and friends. Enlargements of a dozen selected photos were mounted along with quotes and a draft of this article circulated to those present, with each person’s draft highlighting their own words and photos - or in one case - the photos in which they appear. Participants elaborated on the reflections offered previously, suggesting edits and additional detail. Rich discussion then occurred about possible follow-up activities.

Importantly, not all participants in the photovoice exercise are included as co-authors - only those who met the criteria for authorship, namely those who made a substantial contribution to the design and implementation of the project, as well as in the data synthesis and the final paper itself.  Actually, only two participants joined the team after the second photovoice session; other co-authors had already met the criteria for authorship by virtue of their key role in designing and implementing the project.  Indeed, the youngest member of the photovoice group made considerable revisions to the article; the insights from the participants named, especially in the final two photovoice sessions, materially helped shape the paper that was submitted and it would be inconsistent with the ethos of the work for their names to be omitted. All authors approved the submission. (For research in Indigenous communities in Canada, there is a saying: “Nothing about us without us” – and the settler academics on the team certainly wanted to honour this.)

Second, regarding the suggestion of including some pro-pipeline photos or graphics, this was indeed considered at length, but we decided that this would fundamentally change the ethos and focus of the paper.  Including such photos would raise the expectation that we are conducting a systematic visual analysis of messaging by diverse parties in this environmental dispute, and we felt that that would be different paper beyond the scope of our objectives here. Nonetheless, we appreciate the desire for more information on this aspect, so we added a reference to an article that does indeed examine the rhetoric (including language and visuals) used in promoting this pipeline expansion. (Rhetoric Deployed in the Communication Between the National Energy Board and Aboriginal Communities in the Case of the Trans Mountain Pipeline – reference 3).

Third, we thank the reviewer for suggesting that we mention the literature on place attachment; we have now done so, citing both Lewicka 2011 and Scannel and Gifford, 2010, as suggested. Along with the addition of Chandler’s work, as per recommendations of Reviewer 4 regarding cultural continuity, we feel this has indeed strengthened the paper.

Reviewer 4 Report

I absolutely loved this article. The link between land as "place and belonging" is clearly articulated, as is the ongoing impact of colonialist practices in the form of extractivism. The piece is very well referenced, with seminal and recent publications as well as current news/political pieces. I was, however surprised by the absence of Chandler more psychological perspective of "cultural continuity".

My critique is that I thought that the Photovoice methodology could be explained in greater detail, especially since the method was adapted to better address specific cultural realities. That, I think is original in itself and would merit more description. Moreover, the initial question(s) that guided the photo taking/choosing was not clear from the outset. It does become clearer as the results are shared, but it would be easier if they were specifically named.

There is a small typo at line 327-- delete the second "she" 

Author Response

We are delighted that the reviewer loved the article.  We are very happy with it as well.  We also thank the reviewer for the suggestions, to which we respond here:

We have now included reference to Chandler’s work and agree that this work is highly relevant.  Thanks for this.   

Regarding expanding the photovoice methodology, we expanded in 10 places. 

  • We added more to the approach we use, and, as suggested by Reviewer #3, specifically noted the influcne of a place attachment framing on our work. (lines 172-176)
  • We cited one more article on the indigenization of photovoice, Thomspon et al. (reference 22)
  • We expanded on how photovoice was introduced to the community (lines 199-200).
  • We noted (see lines 205-207) “Importantly, researchers purposely provided little direction as to the content of what should be brought forward, leaving it up to participants themselves.” This hopefully helps elaborate on the question raised about the objectives of the exercise – which we address further in the response to the next point below.
  • We elaborated on the set-up for the debriefing: “photos and artifacts that had been sent in by community members or brought by participants were displayed on a table. Alluding to these exhibits,…” (lines 225-226).
  • We also clarified that “The session took the form of a discussion prompted by the photos rather than sequential presentations.” (229-230)
  • We elaborated quite considerably on the issue of authorship(line 245-256) – see Response to Reviewer 3
  • We ensured it was clear who came to the third session (line 256-258)

Regarding the comment that “the initial question(s) that guided the photo taking/choosing was not clear from the outset”, in addition to clarifying this in the Methodology section, as noted above, we revised the final paragraph in Section 1 so it now reads:

The section below contextualises the photovoice project with a brief discussion of why oil pipeline threats and visual storytelling need to be linked with understandings of the sovereign rights of Indigenous peoples. The section thereafter discusses our study methods, designed to support the objective of providing a venue for TWN elders and youth to present experiences and ideas regarding concerns, memories, sustainability issues and values connected to the environment, health and wellbeing, as determined individually and collectively by TWN members. We then explore both the visual stories presented and epistemological issues at play in interpreting key themes and issues that emerged. Finally, we stress the importance of an intergenerational lens of connectedness to nature and sustainability, discussing visual storytelling not just as “counter-narrative” to neocolonial extractivism but also as an invitation into fundamentally different ways of seeing and interacting.

.

We trust the question guiding the photo taking (and in some cases, photo bringing) is now clear.

We also corrected the typo.

Again, we thank the reviewers for their encouragement and helpful comments.